# Mussel-Inspired Surface Functionalization of AEM for Simultaneously Improved Monovalent Anion Selectivity and Antibacterial Property

**DOI:** 10.3390/membranes9030036

**Published:** 2019-03-06

**Authors:** Zhihao Zheng, Pang Xiao, Huimin Ruan, Junbin Liao, Congjie Gao, Bart Van der Bruggen, Jiangnan Shen

**Affiliations:** 1Center for Membrane Separation and Water Science & Technology, Ocean College, Zhejiang University of Technology, Hangzhou 310014, China; zhengzhihaozjut@163.com (Z.Z.); PangX2s@163.com (P.X.); ruanhm@zjut.edu.cn (H.R.); jbliao@zjut.edu.cn (J.L.); gaocj@zjut.edu.cn (C.G.); 2Department of Chemical Engineering, KU Leuven, Celestijnenlaan 200F, B-3001 Leuven, Belgium; bart.vanderbruggen@cit.kuleuven.be

**Keywords:** electrodialysis, antimicrobial, anion exchange membrane, permselectivity

## Abstract

A facile membrane surface modification process for improving permselectivity and antimicrobial property was proposed. A polydopamine (PDA) coating was firstly fabricated on pristine anion exchange membrane (AEM), followed by in situ reduction of Ag without adding any extra reductant. Finally, 2,5-diaminobenzene sulfonic acid (DSA) was grafted onto PDA layer via Michael addition reaction. The as-prepared AEM exhibited improved permselectivity (from 0.60 to 1.43) and effective inhibition of bacterial growth. In addition, the result of the long-term (90-h continuous electrodialysis) test expressed the excellent durability of the modified layer on membrane surface, because the concentration of Cl^−^ and SO_4_^2−^ in diluted chamber fluctuated ~0.024 and 0.030 mol·L^−1^ with no distinct decline. The method described in this work makes the full use of multifunctional PDA layer (polymer-like coating, in situ reduction and post-organic reaction), and a rational design of functional AEM was established for better practical application.

## 1. Introduction

Increasing population nowadays accelerates the crisis of fresh water resources [1]. Seawater, accounting for 97% of the world’s water resources, can provide sustainable clean water after removing salts and other impurities. Because of its high efficiency and low consumption, membrane separation technology is increasingly used in seawater desalination. According to the statistics of International Desalination Association (IDA), more than 60% of the world’s daily output of desalted water is produced by membrane technology [2,3,4].

Electrodialysis (ED), one of the membrane separation technologies driven by electric potential difference on both sides of the membrane [5], is widely used for producing drinking water from brackish water and purification of effluents [4,6,7,8]. Anion exchange membrane (AEM), the core part of electrodialysis system, has the ability to separate anions and cations as the anions in solution would move directionally under the electric field and transfer to the other side of the membrane with the Donnan effect [9]. With the further expansion of application, such as the removal of fluorine/nitrogen from drinking water, brine refining, etc., ED faces a great challenge to treat the complex raw water including the monovalent anions and multivalent anions. The enrichment of multivalent anions in concentrated compartment usually result in the formation of CaSO_4_ precipitation, which decreases the performance of the ED process [10,11]. It is also difficult to remove harmful fluorine and nitrogen anions from drinking water while retaining other divalent anions. AEMs that are capable separating univalent/multivalent anions are in urgent need.

Generally, AEMs with monovalent selectivity are based on sieving and electrostatic repulsion mechanisms [12]. The monovalent selectivity is also influenced by membrane surface morphology and membrane surface hydrophilic–hydrophobic property [13]. Depositing a polyelectrolyte on an AEM with a negative or positive charge is now one of the most promising methods to improve the membrane separation performance [14,15]. Wang et al. [10] and coworkers grafted carboxyl groups onto the membrane surface and then immobilized it with PEI layers. The increase of surface negative charge density and the hydrophobic nature of membrane surface impeded the permeability of SO_4_^2−^. The permselectivity of the modified membrane increased from 0.91 to 2.86. Zhao et al. [16] and coworkers modified an AEM by alternate electrodeposition of polyanions and polycations. The result turned out that the monovalent anion selectivity increased to 2.9 and the separation efficiency increased to 0.28 with nine bilayers due to the accumulated surface negative charge. Surface modification by covalent bonding was also used for better durability. Ding [17] and coworkers prepared AEMs via constructing a covalently cross-linked interface layer by electrodeposition of polyethyleneimine with enhanced stability. Additionally, the monovalent selectivity of the modified AEM increased to 4.29. However, the above-mentioned modified membranes with monovalent/multivalent selectivities had difficulty meeting the demands of situations like complex treatment conditions and storage environments with many pollutants. A multifunctional AEM is now more in-line with actual needs. Therefore, Mulyati [18] and coworkers reported a modified AEM with an odd number of layer-by-layer layers (poly(sodium 4-styrene sulfonate) top layer), which, if above 15, had sufficient monovalent anion selectivity for practical use, and showed high antifouling potential.

Common AEMs with positive charged ion exchange groups inside the membrane easily attract bacteria when treating raw water with microorganisms. And bacteria adhesion is more likely to occur in nonsterile or humid conditions. Therefore, antibacterial activity, as one of the functions of AEM, is more favorable to its practical application. Nowadays, a large number of studies has been done on the antibacterial properties of pressure-driven membranes such as ultrafiltration and nanofiltration [19]. Surface modification of the membrane has a great effect on enhancing the antibacterial property by limiting the adhesion of microorganisms or by killing the bacterial. Xu [20] and coworkers prepared ultrafiltration membranes by blending polysulfone with Ag/Cu_2_O hybrid nanowires; the prepared membrane exhibited enhanced antibacterial performance. The inhibition zone of the membrane increased from 7.9 mm to 28.1 mm with the doping of Ag/Cu_2_O hybrid nanowires. Xie [21] and coworkers fabricated antifouling (organic) and antibacterial membranes by codeposition of dopamine and zwitterionic polymers, followed by incorporating bactericidal silver nanoparticles. However, there are few reports on the antibacterial properties of anion exchange membranes and the studies are insufficient.

Inspired by the universal adhesion of mussel protein, Messersmith et al. [22] found that dopamine can be oxidized in an alkaline aqueous solution and forms a polymer-like coating on a variety of materials with great adhesive strength. Many studies applied the deposition of dopamine on membrane surface for further functionalization because of the abundant active functional groups.

In this study, a novel AEM was fabricated with antibacterial property and monovalent selectivity simultaneously. Dopamine deposition was used as mediated active layer for ulterior multifunctional modification. The antibacterial property was realized via the reduction of Ag nanoparticles [23,24,25]. By utilizing the electrons released by the oxidation of catechol to catecholquinone, the Ag nanoparticles (NPs) were in situ synthesized on the membrane surface without adding any external reducing agents [21,26,27]. 2,5-Diaminobenzenesulfonic acid is a micromolecule with abundant amino, the high reactivity of amino can easily be used in surface modification. The selective functional layer was obtained via the Michael addition reaction and polymerization between the residual active site and amino group, which was induced by ultraviolet cross-linking. Electrodialysis was used to investigate the membranes performance in terms of the selectivity between Cl^−^ and SO_4_^2−^ at a constant current. It turned out that improved rejection of SO_4_^2−^ can be obtained by the introduction of sulfonyl groups. Meanwhile, the Ag nanoparticles on membrane surface significantly enhanced the antibacterial property.

## 2. Experimental

### 2.1. Materials

Dopamine hydrochloride and tris (hydroxymethyl) aminomethane (Aladdin industrial Corporation, Shanghai, China) were used as received. Silver nitrate was purchased from Shanghai SSS Reagent Co., Ltd. 2,5-Diaminobenzenesulfonic acid was purchased from Aladdin Reagent Co. Ltd., Shanghai, China. All the other reagents and solvents were brought from commercial sources and used as received without further purification. Distilled water was used throughout.

Gram negative bacteria (coliform bacteria) is a model biofouling bacterium commonly used to investigate the anti-biofouling property. Luria-Bertani (LB), phosphate-buffered saline (PBS), and agar were purchased from Sinopharm Chemical Reagent Co., Ltd, Shanghai, China.

The membranes used were commercial anion exchange membrane (AEM Type-I) and commercial cation exchange membrane (CEM Type-II) purchased from Fujifilm Corp. Japan; the parameters are shown in Table 1. The area resistance was measured with 0.5 M NaCl solution.

### 2.2. Membrane Modification

#### 2.2.1. Synthesis of Silver Nanoparticles Chelated Dopamine Coating onto Membrane Surface

The commercial original AEM was alternately immersed in the NaOH solution (0.2 M) and HCl solution (0.2 M) for 30 min to remove the impurities. Dopamine solution with a concentration of 2 mg/1 mL was prepared by dissolving dopamine in tris (hydroxymethyl) aminomethane-HCl (Tris-HCl) buffer solutions (pH = 8.5). Subsequently, the prepared solution was added into a custom-designed experimental setup assembled with a certain area of pristine AEM (shown in Figure 1). The dopamine solution darkens over time and the polymerization of dopamine lasted 4 h. The prepared membranes were rinsed with deionized water several times and the membranes were termed DA/AEM. 

Then, the above prepared membranes (DA/AEM) were immersed single-sided with AgNO_3_ (0.02M) solution for 12 h. In this experiment, silver ions were supposed to be reduced into metallic silver and grown into silver nanoparticles without adding any external reducing agents.

The membranes incorporated with Ag were then rinsed with deionized water and were named DA/Ag/AEM.

#### 2.2.2. Grafting of Sulfonic Functional Groups on Prepared Membranes

The AEMs with PDA deposition were supposed to be further surface-modified and functionalized via secondary treatments, because of their robust reactivity. To endow the membranes with permselectivity, 2,5-diaminobenzene sulfonic acid (DSA), which is rich in amino functional groups, was used as a reactive carrier containing sulfonic acid, and can react with polydopamine by Michael addition [28].

Herein, the prepared DA/Ag/AEM (DA/AEM) was loaded in a lab designed setup (seen in Figure 2) and immersed with 2.46 mg/mL DSA/Tris-HCl buffer solution. The device with membrane immersed was exposed to Ultraviolet (UV) light (260 nm, 7.5 mW·cm^−2^, HgXe lamp) in UV cross-linkers for 12 h, helping induce the grafting reaction between PDA and DSA. The prepared membranes were then rinsed with deionized water several times, and the membranes were named DA/Ag/DSA/AEM (DA/DSA/Ag/AEM).

### 2.3. FTIR Characterization and X-ray Photoelectron Spectroscopy (XPS)

The chemical composition and structure of silver-loaded and -unloaded membranes were determined by XPS (KratosAXIS Ultra DLD, Kyoto, Japan) and FTIR spectra (ATR-FTIR, Nicolet6700, Thermofisher, New York, NY, USA). The prepared membranes were scanned from 400 to 4000 cm^−1^ by Attenuated Total Reflectance.

Attenuated Total Reflec Fourier Transform Infrared Spectra (ATR-FTIR) and X-ray photoelectron spectroscopy (XPS) were utilized to detect the elemental composition of different type of membranes. Membrane samples were dried under vacuum at 25 °C before tests.

### 2.4. Electrochemical Characterization of Modified AEMs

#### 2.4.1. Membrane Surface Resistance and Ion Exchange Capability

In order to measure the surface resistance of the membranes, the pristine and modified membranes were immersed in a 0.5 M NaCl solution or a 0.5 M Na_2_SO_4_ solution to reach the equilibrium of ion-exchange adsorption prior to measurement. Membrane surface resistance measurements were carried out in a home designed setup (seen in Figure 3) with the solution of 0.5 M NaCl or 0.5 M Na_2_SO_4_ solution at ambient temperature, the effective membrane surface area was 7.065 cm^2^. Ag/AgCl electrodes were utilized to determine the potential difference between the two sides of membranes under the condition of a constant current. In order to reduce the concentration polarization and diffusion effect of the solution in the feed chamber, the solutions in the two chambers are interlinked and the continuous agitation is replaced by a pump during resistance measurement process. The membrane surface resistance were calculated according to the following equation.
(1)$$R_{n}=\frac{U-U_{0}}{I}\times S$$
where *R_n_* represents the surface resistance of membranes expressed in Ω·cm^2^, *U* represents the voltage value of the membrane and *U*_0_ represents the voltage of blank expressed in V, and *I* represents the constant current through the membrane and insure the current at 0.04 A.

After being dried in vacuum at 50 °C overnight, the AEM samples were then weighed and immersed in 1 mol/L NaCl solution for 24 h to transform the exchangeable anion group in the membrane into Cl^−^ anions. Afterward, the samples were rinsed with deionized water completely, and the leacheate was tested with a 0.1 mol/L AgNO_3_ solution to make sure there was no AgCl precipitation observed. Next, the samples were immersed in 50 mL 0.5 mol/L Na_2_SO_4_ solution for 24 h to exchange all the Cl^−^ anions in membrane into solution and the resultant immersion solution was finally titrated by 0.01 mol/L AgNO_3_ solution. The above operation was repeat for 3 times to obtain the average data and the titration process was carried out via Automatic Potentiometric Titrator (METTLER TOLEDO T50, Zurich, Switzerland). The IEC was calculated by the formula
(2)$$IEC=\frac{n_{Cl^{-}}}{m_{d}}$$
where $n_{Cl^{-}}$ represents the amount of substance of chloride ions expressed in mmol and $m_{d^{}}$ represents the weight of dry membrane expressed in g.

#### 2.4.2. ζ-Potential

The electrical properties on the surface of different types of membranes were investigated by a Zeta-potential electrokinetic analyzer (SurPASS, Anton Paar, Glaz, Austria) with 1 mM KCl as the electrolyte solution. The pH dependence of surface zeta potential was investigated via adjusting the pH by using NaOH and HCl solutions.

### 2.5. Monovalent Anions Selectivity Measurement

The selectivity between divalent and monovalent anions of membrane was measured in a four-cell ED apparatus (shown in Figure 4). The volume of each compartment is 100 mL and the effective area of the membrane is ~19.625 cm^2^. The modified membrane was clamped in the middle of the four compartments, and the side of membrane’s active layer was directed towards the cathode. The compartments near the electrodes were divided by two commercial cation exchange membranes for inhibiting the leakage of anions in electrode solution to the dilute compartment. The 0.05 M Na_2_SO_4_ and 0.05 M NaCl mixed solution was used as feed solution in the middle of the two compartments, and a 0.2 M Na_2_SO_4_ solution was used as electrode solution. The electrodialysis experiment was carried out at the current density of 5.1 mA·cm^−2^ for 2 h. The concentration of Cl^−^ and SO_4_^2−^ in the dilute compartment were measured by an anion chromatography (Thermo Fisher ICS-1100, Thermofisher, New York, NY, USA) at room temperature every 30 min. The permselectivity of membranes between Cl^−^ and SO_4_^2−^ was calculated by the following equation.
(3)$$P_{{\mathrm{SO}}_{4}^{2-}}^{{\mathrm{Cl}}^{-}}=\frac{t_{Cl^{-}}/t_{SO_{4}^{2-}}}{c_{Cl^{-}}/c_{SO_{4}^{2-}}}=\frac{J_{Cl^{-}}\times c_{SO_{4}^{2-}}}{J_{SO_{4}^{2-}}\times c_{Cl^{-}}}\times 100\%$$
where *t_i_* represents the transport number of the ions in the membrane, *J_i_* represents the flux of the target anion through the membrane expressed in mol·m^−2^·s^−1^, and *c* represents the concentration of anions in the dilute compartment expressed in M. The flux of ions was obtained from the change in concentration of the ions on the dilute side:(4)$$J_{i}=\frac{V\times \frac{dc_{i}}{dt}}{A}$$
where *V* is the volume of the electrolyte solution in the dilute compartment, which was 100 mL, and *A* is the active area of the membranes, which was 19.625 cm^2^.

### 2.6. Antibacterial Test of Membranes

#### 2.6.1. Antibacterial Activity Test

The antibacterial activity of the membranes was tested toward the model bacterial: Gram-negative *Escherichia*
*coli* (abbreviated as *E.*
*coli*). The inhibition zone method was utilized to determine the antibacterial activity, and membranes were sterilized before the test. Then, membranes were placed on the top of Luria Bertani (LB) agar plates (containing 10 g/L peptone, 5 g/L yeast extract, 10 g/L sodium chloride, and 16 g/L agar at a pH of 7.0) uniformly cultured with *Escherichia coli* bacteria at the concentration of 10^6^ cfu·mL^−1^ and incubated at 37 °C overnight. The diameters of inhibition zones in the sample disks were measured and recorded using a digital camera.

#### 2.6.2. Bacterial Suspension Test

The bacterial suspension test was carried out to describe the antibacterial activity of the modified membranes. All of the membrane samples were disinfected by ultraviolet radiation for 30 min before test. Then, membranes were immersed in 10 mL of *E. coli* bacterial suspension (10^6^ cfu·mL^−1^) and incubated at 37 °C with a stirring speed of 250 rpm to grow the bacterial overnight. To test the antibacterial efficiency, the prepared membranes were taken out and washed with ultrapure water followed by dealing with 3% (*v/v*) glutaraldehyde for 5 h at 4 °C [29]. At last, the surface morphologies of the pristine and modified membranes were observed by scanning electron microscopy (SEM) (SU8010 Hitachi, Tokyo, Japan) with an acceleration voltage of 5 kV. All samples were fixed on a SEM sample holder with double-sided conductive adhesive and then were sputter-coated with 10 nm of gold before imaging.

## 3. Result and Discussion

### 3.1. Surface Characterization of the Membrane Surfaces

Attenuated total reflectance-Fourier transform infrared spectroscopy (ATR-FTIR) was utilized to identify the chemical compositions of membrane surfaces (see in Figure 5). After the ultraviolet-induced Michael addition, DSA was easily grafted onto the membrane surface with a polydopamine layer. The characteristic peak, which occurred at ~1022 cm^−1^, corresponds to the symmetric stretching vibrations of sulfonyl (–SO_3_–) groups, indicating that the successful functionalization of the dopamine onto membrane surface with DSA. In addition, the appearance of the absorbance band at 1182 cm^−1^ can be assigned to the symmetric stretching vibrations of the –S–O band [16], which further proved the grafting of sulfonyl groups.

The surface chemical composition of pristine membrane and the chemical state of silver on membrane surface of modified membrane were further investigated by XPS and XRD.

The XPS spectrum of the in situ reduced silver-loaded membrane is shown in Figure 6a,b. It can be seen that Ag is present at the DA/Ag modified membrane, and that Ag and S are present at DA/Ag/DSA modified membrane. The S is attributed to the Michael addition reaction of 2,5-Diaminobenzene sulfonic acid. Characteristic peaks of Ag are attributed to the Ag nanoparticles, and the binding energies of the doublets are found to be 367.95 eV (Ag3d5/2) and 373.85 eV (Ag3d3/2) in the XPS narrow spectrum, representing the characteristics of metallic Ag [30,31]. Figure 6c shows the XRD patterns of the pristine membrane and Ag nanoparticle immobilized membrane. It can be seen that the strong characteristic peak located at 2θ of 38.2. Weak characteristic peaks located at 2θ of 44.3°, 64.6°, and 77.7 are assigned to the (111), (200), (220), (311) planes of the cubic structure of metallic Ag, respectively [31]. These characteristic peaks indicate that the silver particles on membrane surface are in the metallic state.

The in situ reduction mechanism of silver ions were shown in step (2) in graphic abstract, the silver ions in solution were firstly chelated by the hydroxyl groups on the polydopamine, and then the silver ions were in situ reduced to silver atoms via accepting the electrons released by the simultaneously oxidation of catechol to catecholquione structures. Due to the equilibrium state of catechol and catecholquione groups [32], the reduction of silver ions was proceeded continuously without adding extra reducing agent.

### 3.2. Membrane Surface Resistance and Ion Exchange Capability

Surface modification was an effective way to exchange physicochemical property of membrane surface, which was convenient to improve some specific performance. However, surface modification increases the surface membrane resistance, resulting in a decrease of the current efficiency [33]. The surface resistance of modified membrane and pristine membrane are listed in Table 2. Among them, the surface resistance of pristine membrane was 1.03 ± 0.019 Ω·cm^2^, which was 0.24 Ω·cm^2^ higher after DA modification. It could be explained that the deposited DA layer increased the thickness of the integral membrane and endowed the membrane surface with reinforced electronegativity, as shown in Figure 7. The surface resistance of DA/Ag membrane was 1.28 ± 0.01 Ω·cm^2^, which was almost the same with DA membrane (1.27 ± 0.02 Ω·cm^2^). The surface resistance of DA/Ag/DSA membrane was 1.49 ± 0.02 Ω·cm^2^, which was lower than DA/DSA membrane (1.67 ± 0.02 Ω·cm^2^). The reduced silver atoms could be bonded on the N-site and O-site in the polydopamine layer, which means there were less reactive sites left for DSA. Compared with the DA/DSA membrane, the less-grafted DSA around the Ag nanoparticles on the DA/Ag/DSA membrane surface resulted in the more leaked chloride ions passing through the selective layer. In other words, chloride ions passed through the membrane more easily and expressed low surface resistance. In addition, the surface resistance of membranes in 0.5 M Na_2_SO_4_ solution expressed a similar trend but higher value. This phenomenon can be explained by the fact that sulfate ions were subjected with greater resistance than chloride ions through the membrane.

The ion exchange capability (IEC) of the membrane declined after surface modification. As shown in Table 2, the IEC of pristine AEM was 1.71 mmol·g^−1^, and the IEC of the modified AEM slightly decreased to 1.67 mmol·g^−1^ (DA/AEM), 1.55 mmol·g^−1^ (DA /DSA/AEM), 1.66 mmol·g^−1^ (DA/Ag/AEM), and 1.54 mmol·g^−1^ (DA/Ag/DSA/AEM). This is due to the electrostatic neutralization between the sulfonyl groups and quaternary ammonium groups. With the partly blocked transfer sites, the decreased ion exchange sites of the modified AEM resulted in the decline of IEC.

### 3.3. ζ-Potential of Membrane Surface

The charge property surrounding the membrane surface could be changed after surface modification, which influences the performance of as-prepared AEM during the ED process. For instance, anions can effectively enter the membrane via the Donnan effect [11] and transfer to the other side of membrane under direct current field. AEM modified with electronegative materials would partly inhibit multivalent anions due to the electrostatic repulsion. The ζ-potential was performed to describe the electrical surface charges of the membrane surface via the inversion of membrane surface charge with mutative surroundings [16]. Figure 7 shows the ζ-potentials at various pH values of the (a) pristine AEM, (b) DA/Ag/DSA AEM, and (c) DA/DSA AEM. With increasing pH value, both DA/DSA and DA/Ag/DSA AEM show a more negative charge. The introduction of negative sulfonyl groups and the deprotonation of phenolic hydroxyl on polydopamine, the modified layers exhibit more characteristics of anionic polyelectrolytes. It can be inferred that DA/Ag/DSA and DA/DSA AEM have the potential to express higher permselectivity than the pristine AEM. In addition, the DA/DSA membrane expressed higher zeta potential than DA/Ag/DSA at low pH. This phenomenon can be explained by the loading of Ag NPs on the membrane surface, which means that the surface of polydopamine was partly covered with Ag NPs. The less-grafted DSA and partly covered polydopamine supplied fewer amino groups for protonation at low pH, which resulted in lower potential of DA/Ag/DSA than DA/Ag at low pH.

### 3.4. Monovalent Anion Selectivity

Permselectivity is a reference indicating the transport of monovalent and multivalent anions through the membrane. Cl^−^ and SO_4_^2−^ were chosen as target anions in the diluted compartment and the permselectivity was calculated and shown in Figure 8.

The temporal evolution of transport number ratios changed in the diluted compartment was indicated by permselectivity. All of the membranes exhibited improved permselectivity after modification. As shown in Figure 8, the permselectivity of pristine AEM fluctuated approximately 0.6 during the 120 min ED process. DA AEM and DA/Ag AEM exhibited a similar performance of permselectivity under the Cl^−^/SO_4_^2−^ system. Compared with pristine AEM, the permselectivity of DA AEM and DA/Ag AEM increased to 0.98 and 1.0, respectively. The deprotonation of polydopamine partly inhibits the migration of anions under electrostatic repulsion; meanwhile, the deposition of a macromolecule could impede the anions with bigger hydrated ionic radius by forming a denser surface layer. The permselectivity of DA/DSA AEM and DA/Ag/DSA AEM were further enhanced by the introduction of sulfonyl groups, which increased to 1.42 and 1.43, respectively. The charged property of membrane surface were shown in Figure 7. The in situ reduction of silver on the membrane surface had a negligible effect on the permselectivity.

To evaluate the duration performance of modified membranes, a continuous electrodialysis was operated with a total time of 90 h. The long term ED test lasted for 90 h with the same DA/Ag/DSA AEM under a current density of 5.1 mA·cm^−2^. During the 90-hour electrodialysis, the tested membrane was never replaced. The ED device was rinsed and filled with new feed solution (0.05 M Cl^−^ and SO_4_^2^^−^) every two hours [15,34], and the samples were taken and analyzed for anion content at the end of every two hours. As shown in Figure 9, the changes of concentration (Cl^−^ and SO_4_^2−^ anions) during 90 h ED process have expressed acceptable limits: both Cl^−^ and SO_4_^2−^ anions were centered at 0.024 and 0.030 mol·L^−1^_­_, respectively. The slight change of concentration indicates the strong adherent strength of the PDA layer which provides a good modifying medium and consolidates the subsequent modification. Thus, the prepared AEMs possess good stability for the ED process.

### 3.5. Antibacterial Test

Reduced Ag NPs on membrane surface was supposed to endow modified AEM with antibacterial property via multi-interactions with the bacteria, including proteins, DNA, and the bacterial cell wall [23,35].

The bactericidal activity of modified AEM was firstly tested via a bacterial inhibition zone toward Gram-negative *E. coli*. As can be observed in Figure 10, no obvious inhibition zone was observed around (a) pristine AEM, (b) DA AEM, and (c) DA/DSA AEM due to the lack of limitation of bacteria growth. The result indicates the non-antibacterial property to *E. coli* for these membranes. After the in situ reduction of Ag, the modified membrane showed a distinct inhibition effect on *E. coli*. The result was shown in Figure 10, the diameter of inhibition zone toward *E. coli* of (d) DA/Ag AEM reached 10.6 mm, while the diameter of inhibition zone of (e) DA/Ag/DSA AEM reached 11.3 mm. The result indicates that the presence of the Ag nanoparticles on membrane surface significantly enhance the antimicrobial property and exhibit excellent inhibition capacity of membrane.

The bacteria growth test on the membrane surface was carried out to evaluate bacterial growth activity under an aqueous solution system in the presence of a modified membrane. The fresh bacterial suspension with a concentration of 10^6^ cfu·mL^−1^ replaced the bacterial medium and cocultured with the modified AEM for 12 h. The morphology of (a) pristine AEM, (b) DA AEM, (c) DA/DSA AEM, (d) DA/Ag AEM, and (e) DA/Ag/DSA AEM without *E. coli* was shown in Figure 11. No bacteria was observed on pristine AEM and modified AEM, while there were a few polymer-like structures on (b) DA AEM and (c) DA/DSA AEM, they may be ascribed to the polymerization of dopamine and the grafting of DSA. Compared with Figure 6, Ag nanoparticles were confirmed on the surface of (d) DA/Ag AEM and (e) DA/Ag/DSA AEM. The morphology of bacteria on the surface of the tested membrane cocultured with bacterial suspension is shown in Figure 12. A similar conclusion could also be drawn that lots of bacteria on the surface of (a) pristine AEM, (b) DA AEM, and (c) DA/DSA AEM were observed clearly, while almost no bacteria exist on the surface of (d) DA/Ag AEM and (e) DA/Ag/DSA AEM. This phenomenon indicates that the Ag nanoparticles on the surface of (d) DA/Ag AEM and (e) DA/Ag/DSA AEM endowed these modified AEMs with excellent antimicrobial property.

## 4. Conclusions

In this study, dual-functional (monovalent selectivity and antibacterial property) membrane surfaces were successfully fabricated using a multifunctional PDA layer. The improved selectivity reached 1.43 of DA/Ag/DSA AEM was prepared using the Michael addition reaction between the catechol groups of PDA and the amino groups of 2,5-Diaminobenzene sulfonic acid, and the Ag nanoparticles were in situ reduced on a membrane surface for enhanced microbial property. Meanwhile, the modified layer on membrane surface expressed good durability during the 90-h continuous electrodialysis process and the permselectivity showed no obvious change. The easy synthesis procedure involved during the proposed modification process and the improved performance indicates the potential of modified membranes in the separation of monovalent and multivalent anions, and the better antimicrobial property expands the applied range for broader application conditions.

## Figures and Tables

**Figure 1 membranes-09-00036-f001:**
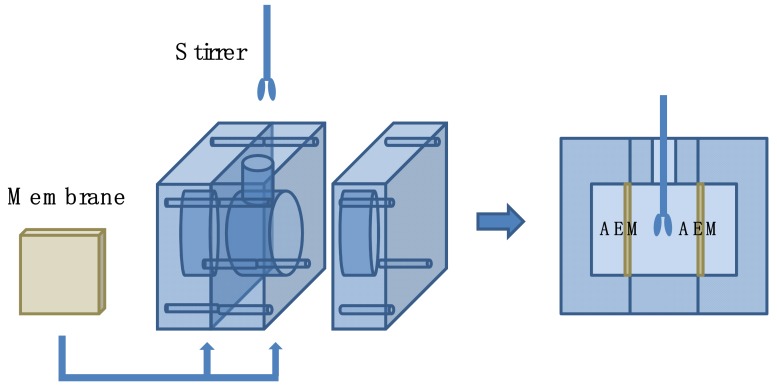
A custom-designed experimental setup for modification of anion exchange membranes (AEMs).

**Figure 2 membranes-09-00036-f002:**
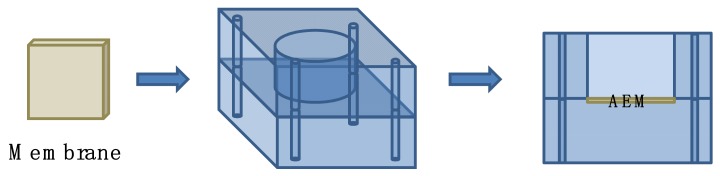
A lab designed setup for membrane modification.

**Figure 3 membranes-09-00036-f003:**
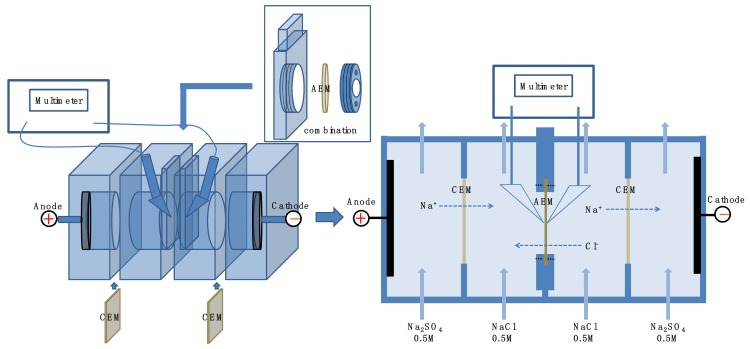
Schematic drawing of a four-compartment device for membrane surface area resistance measurement.

**Figure 4 membranes-09-00036-f004:**
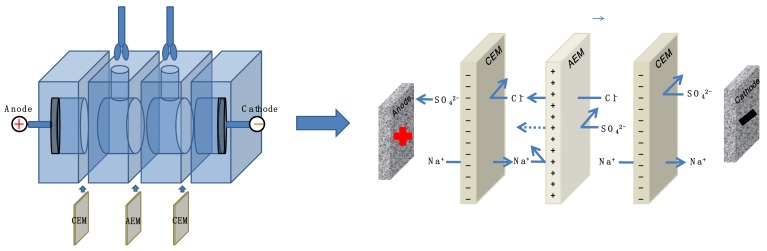
Schematic drawing of a four-compartment device for the monovalent anion selectivity measurement.

**Figure 5 membranes-09-00036-f005:**
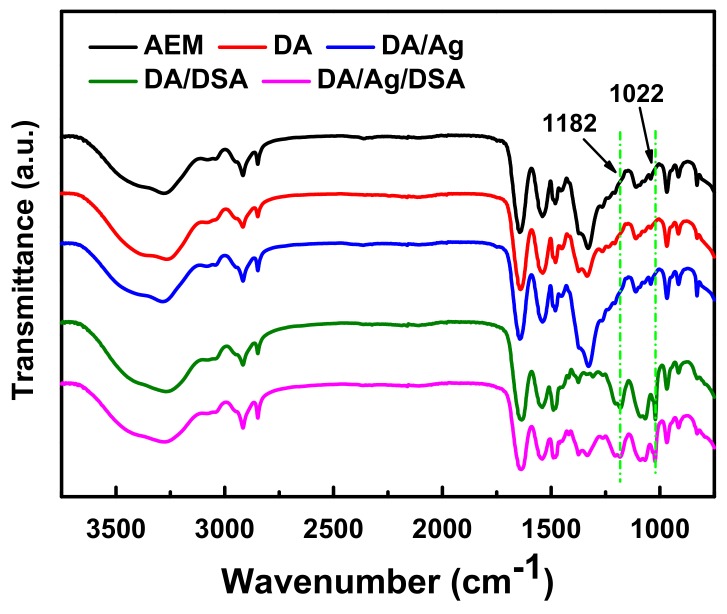
Fourier transform infrared (FTIR) spectra of the pristine AEM and the modified AEMs.

**Figure 6 membranes-09-00036-f006:**
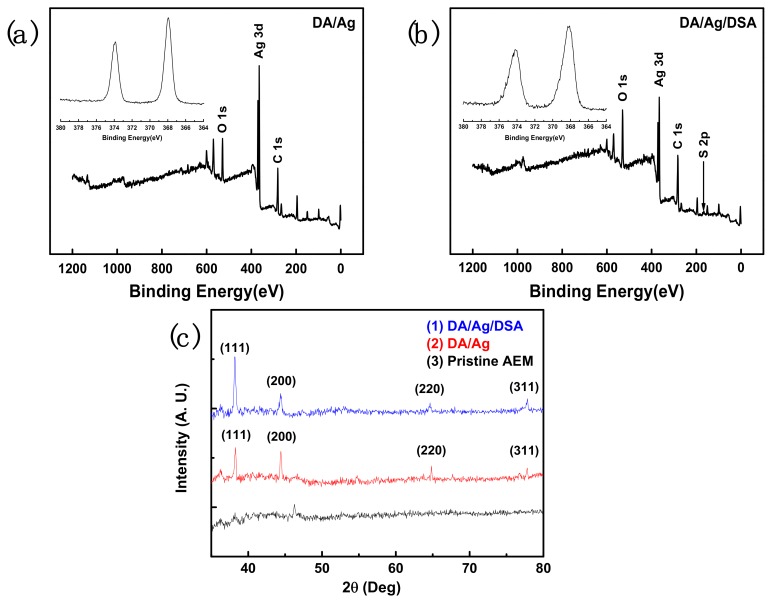
XPS spectra of (**a**) DA/Ag/AEM and (**b**) DA/Ag/DSA/AEM. (**c**) XRD spectra of pristine AEM and modified AEM.

**Figure 7 membranes-09-00036-f007:**
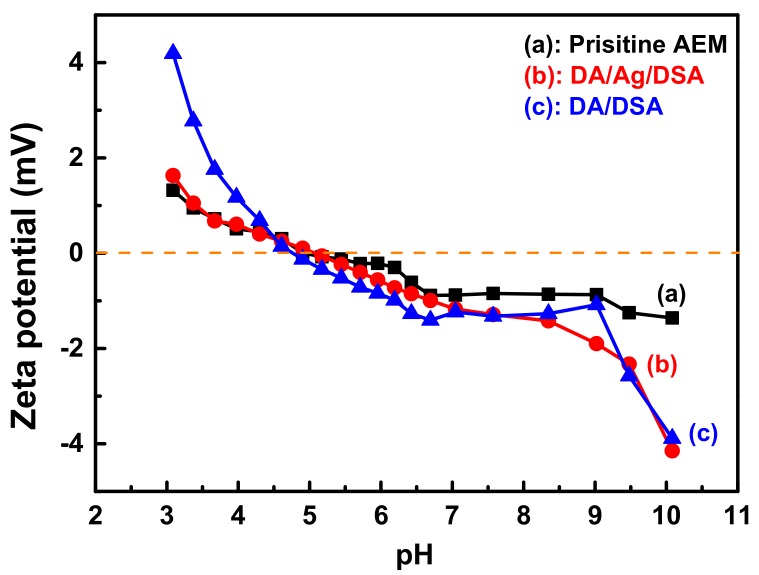
The ζ-potentials of (**a**) the pristine AEM, (**b**) the DA/Ag/DSA AEM, and (**c**) the DA/DSA AEM at various pH values.

**Figure 8 membranes-09-00036-f008:**
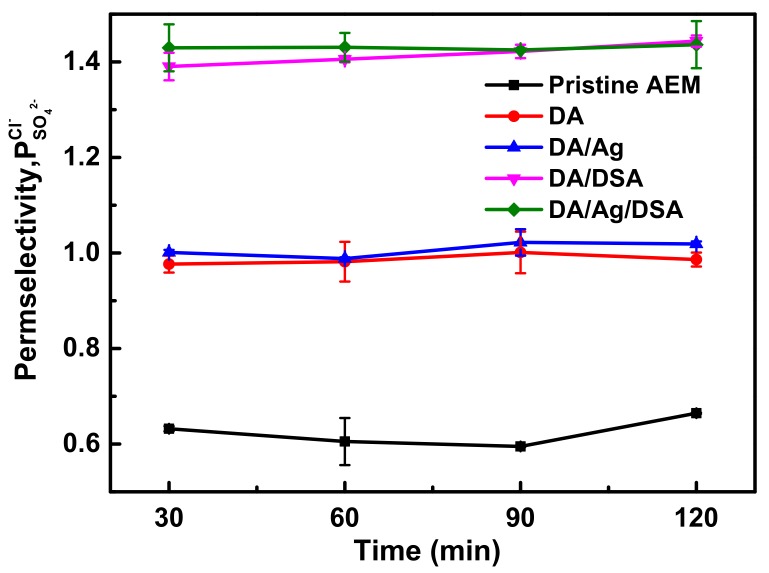
Permselectivity of pristine AEM, DA AEM, DA/Ag AEM, DA/DSA AEM, and DA/Ag/DSA AEMs in the ED process under the system of Cl^−^/SO_4_^2−^.

**Figure 9 membranes-09-00036-f009:**
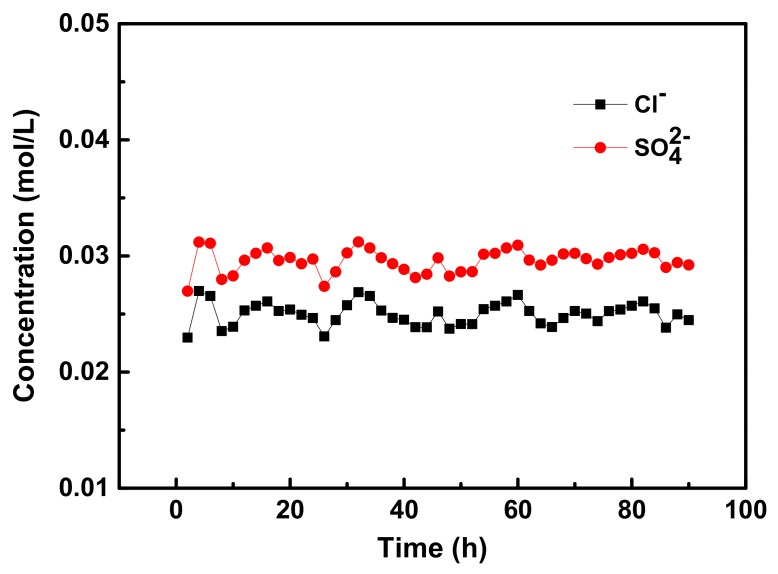
The stability test (Cl^−^ and SO_4_^2−^ system) of the DA/Ag/DSA AEM under the ED process.

**Figure 10 membranes-09-00036-f010:**
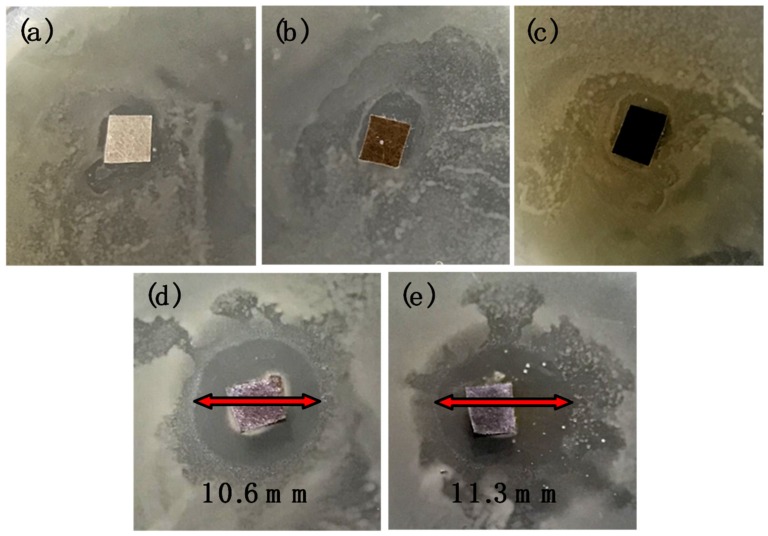
Inhibition zone test towards *E. coli* of (**a**) pristine AEM, (**b**) DA AEM, (**c**) DA/DSA AEM, (**d**) DA/Ag AEM, and (**e**) DA/Ag/DSA AEM.

**Figure 11 membranes-09-00036-f011:**
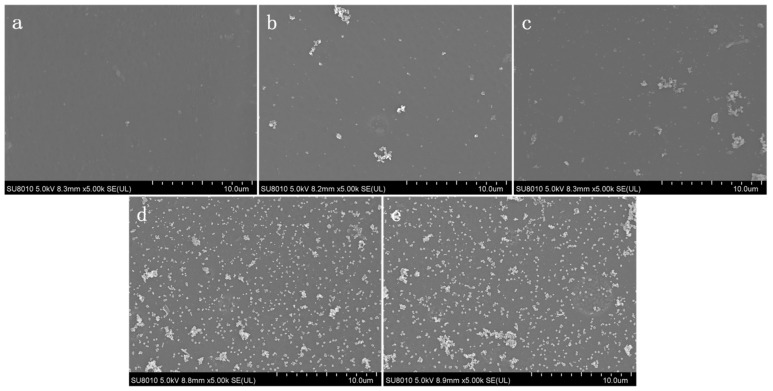
Morphology of (**a**) pristine AEM, (**b**) DA AEM, (**c**) DA/DSA AEM, (**d**) DA/Ag AEM, and (**e**) DA/Ag/DSA AEM without *E. coli*.

**Figure 12 membranes-09-00036-f012:**
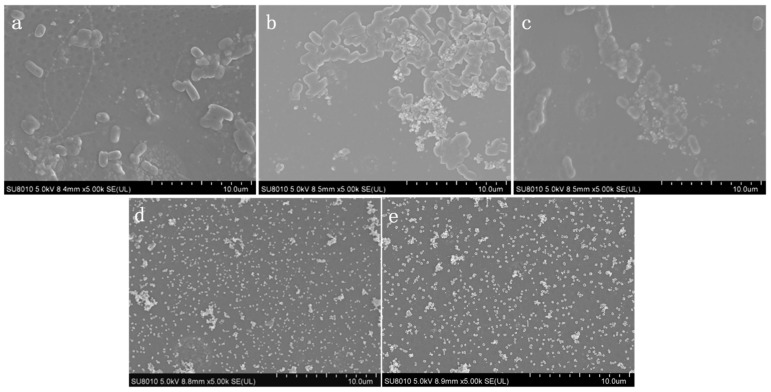
Morphology of *E. coli* on (**a**) pristine AEM, (**b**) DA AEM, (**c**) DA/DSA AEM, (**d**) DA/Ag AEM, and (**e**) DA/Ag/DSA AEM.

**Table 1 membranes-09-00036-t001:** Characteristics of the anion and cation exchange membranes (commercial data).

Membrane Type	Thickness (μm)	Area Resistance (Ω·cm^2^)	pH Stability	Functional Group
Homogeneous (AEM-Type I)	125	1.3	2–10	Quaternary amino group
Homogeneous (CEM-Type II)	135	2.7	4–12	Sulfonic group

**Table 2 membranes-09-00036-t002:** Surface resistance and the ion exchange capability of both the pristine and modified AEM.

Type	Area Resistance (Ω·cm^2^)		IEC (mmol·g^−1^)
	in 0.5 M NaCl Solution	in 0.5 M Na_2_SO_4_ Solution	
Pristine AEM	1.03 ± 0.02	3.09 ± 0.03	1.71 ± 0.02
DA/AEM	1.27 ± 0.02	3.79 ± 0.06	1.67 ± 0.02
DA/DSA/AEM	1.67 ± 0.03	5.15 ± 0.05	1.55 ± 0.01
DA/Ag/AEM	1.28 ± 0.01	3.92 ± 0.04	1.66 ± 0.01
DA/Ag/DSA/AEM	1.49 ± 0.02	4.85 ± 0.05	1.54 ± 0.02

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
