# Peer review of "Mussel-Inspired Surface Functionalization of AEM for Simultaneously Improved Monovalent Anion Selectivity and Antibacterial Property"

_membranes, 2019, doi:10.3390/membranes9030036_

Round 1

Reviewer 1 Report

The manuscript is interesting, however, before publication it should be substantially improved. E.g. the selectivity was determined only for one mixture - 0.05 M Na2SO4 + 0.05 M NaCl. These concentrations are low; it would be useful to perform such experiment for 0.5 M to show how the increased ionic strength influences the monovalent anion selectivity. Regarding the ED experiment (section 2.5, Fig.4), it seems that there was neither the circulation of solutions nor stirring in the ED module. If it is true then the experiments should be repeated.

Some other comments:

1. Eq.(2) – n(H+) – where are H+ ions from??

2. Section 2.5: „The concentration of Cl- and SO42- in the dilute compartment were measured … after every 20 min.” – really? According to Fig. 8 it would be every 30 min.

3. Eq.(3): comparing to t, J includes also the diffusion.

4. Eq.(4) - how was dc/dt determined?

5. Section 3.4: please, give some details on "continuous electrodialysis".

… etc.

 The most important lack of this manuscript is only a one concentration point for which the anion selectivity was determined. One point is nothing, especially that it was a dilute solution.
Secondly, the aim of paper is poorly defined; the Authors do not refer to their previous paper:
H.M. Ruan, Z.H. Zheng, J.F. Pan, C.J. Gao, B. Van der Bruggen, J.N. Shen, Mussel-inspired sulfonated polydopamine coating on anion exchange membrane for improving permselectivity and anti-fouling property, J Membrane Sci, 550 (2018) 427-435.
What is new in their present manuscript comparing to that publication?

Apart of that, important details in the description of experiments are lacking. They should describe the experimental part more precisely.

Author Response

Point-to-Point Response to Reviewers' Comments:

Re:Manuscript ID membranes-435784

Reviewer #1:

Q1. The manuscript is interesting, however, before publication it should be substantially improved. E.g. the selectivity was determined only for one mixture - 0.05 M Na2SO4 + 0.05 M NaCl. These concentrations are low; it would be useful to perform such experiment for 0.5 M to show how the increased ionic strength influences the monovalent anion selectivity. Regarding the ED experiment (section 2.5, Fig.4), it seems that there was neither the circulation of solutions nor stirring in the ED module. If it is true then the experiments should be repeated.

R: Thank you for your professional suggestion. The electrodialysis experiment is carried out with continuous stirring. And Fig. 4 has been modified in the revised manuscript. The concentration we used is appropriate for our device and pipe line, in addition, the low concentration can avoid the occurrence of concentration polarization. we will order new instruments and explore the selectivity at different concentrations in subsequent experiments.

Fig. 4 Schematic drawing of a four-compartment device for Monovalent Anions Selectivity Measurement.

Q2. Eq.(2) – n(H+) – where are H+ ions from??

R: Thank you for pointing out our mistake, the H+ in equation 2 should be replaced by Cl- in this case.

Q3. Section 2.5: „The concentration of Cl- and SO42- in the dilute compartment were measured … after every 20 min.” – really? According to Fig. 8 it would be every 30 min.

R: Thank you for pointing out our carelessness.

Q4. Eq.(3): comparing to t, J includes also the diffusion.

R: Because of the low concentration and continuous stirring, the diffusion phenomenon is ignored here.

Q5. Eq.(4) - how was dc/dt determined?

R: dc/dt means the change in concentration per unit time.

Q6. Section 3.4: please, give some details on "continuous electrodialysis".

R: More detailed explanation has been added to the manuscript. The solution on the side of the diluted chamber is exchanged with new feed solution (0.05 M Cl-/SO42-) every two hours and the samples were taken and analyzed for anion content at the end of every two hours.

Q7. The most important lack of this manuscript is only a one concentration point for which the anion selectivity was determined. One point is nothing, especially that it was a dilute solution.

Secondly, the aim of paper is poorly defined; the Authors do not refer to their previous paper:

H.M. Ruan, Z.H. Zheng, J.F. Pan, C.J. Gao, B. Van der Bruggen, J.N. Shen, Mussel-inspired sulfonated polydopamine coating on anion exchange membrane for improving permselectivity and anti-fouling property, J Membrane Sci, 550 (2018) 427-435.

What is new in their present manuscript comparing to that publication?

R: The permselectivity was shown in Fig.8. The previous work focused on the pre-modification of dopamine followed by polymerization on the membrane surface to improve selectivity and resistance to organic fouling. In this paper, a polydopamine layer was deposited on the membrane surface first and then a post-modification was established to improve selectivity and antibacterial activity. The methods of modification and the emphases are different.

Reviewer 2 Report

The paper reports the surface modification of anion-exchange membranes with successive layers of polydopamine and silver. The aim of the study is to impart antibacterial properties and selectivity for monovalent anions.  The modification of membranes with polydopamine has become trendy in the recent times due to its simplicity. Nonetheless, there is some information which is not clearly explained in the manuscript. In my opinion, the connection between the coatings and the change in properties of the membranes is not sufficiently explained. I list more detailed comments on the manuscript below:

1.       The title of the manuscript is somehow confusing. It gives more importance to the inspiration on mussel secretions than other, in my opinion, more relevant aspects of the work. It does not express which kind of permselectivity is improved (for monovalent anions) and it leads to the idea that the antibacterial property is caused by the “mussel inspired surface functionalization” instead of silver reduced on the membranes.

2.       The article should be revised by an English native speaker. Here I report some of the parts in the text which are confusing:

-          Abstract needs exhaustive corrections: PDA coating was fabricated, Michael addition reaction was proceed, diluted department, the method described in this work making… (no verb in this sentence)

-          l65. Would easily attracts bacteria […] Killing the bacterial

-          l130. Riching

-          l215. dealing with glutaraldehyde solution…

-          (l256) it could be explained by the deposited…

-          (l261) the in situ synthesized Ag NPs was endowed the membrane…,

-          (l266) due to the electrostatic charge neutralization between sulfonyl groups,

-          (l274) anions can effectively entering into membrane by Donnan…,

3.       As mentioned before, the authors should clearly emphasize which type of permselectivity was improved: the readers could think that they refer to the selectivity towards anions, as compared to cations.

4.       Why do the authors focus on antibacterial growth? Are problems reported with ED processes? Could the authors support the motivation of this type of modification with references reporting problems of anion-exchange membranes with bacteria?

5.       L 92. What do the authors mean by improved salt rejection? Anion-exchange membranes are expected to let ions pass through, while avoid, at certain level the transport of water. If the transport of Cl- ions is promoted, salt rejection should not decrease.

6.       Coating with Ag has been widely used to promote antibacterial properties in membranes. However, it can also become a critical issue in case Ag is not properly immobilized and it is released, leading to toxic effects on consumers. Did the authors check the stability of Ag coatings?

7.       Coating with Ag is implemented to endow the membranes with antibacterial properties, wile DMA coating is applied to increase monovalent anion permselectivity. Could the authors explain in more detail which is the purpose of coating with polydopamine? I guess this is an important point, according to the reference to the mussel inspired modification mentioned in the title. However, from reading the text it is not clearly explained.

8.       L155. Which constant current was applied? Was it below the corresponding limiting current?

9.       L224. When the authors say, “the characteristic peak around 1022 cm-2 corresponds to the symmetric stretching…”. Do they mean a negative peak?

10.   On line 261, I do not understand the increase in tunnel conduction effect and field emission conductive effect. Is the ionic conductivity taking place through the ED cell (electrolyte + membrane) of the same character of electric conductivity? How are both types of conductivities related? Is there any reaction occurring in the Ag NPs deposited on the membrane? As far as I know ion exchange membranes are required to have high ionic conductivity but, unless they are used as electrodes, they are not required to have electric conductivity.

11.   Please, explain what diameter of inhibition is. This term may not be familiar to many membrane scientists.

12.   It is reported that the successive coatings induce an increase in resistance of the membranes. However, the resistance was calculated using NaCl solutions, without divalent ions. It has been reported in previous studies, that an increase in selectivity for monovalent ions occurs at the cost of an increase in electrical resistance of the membranes. However, the increase in resistance is usually higher when treating solutions containing divalent ions. (Check for example F. Roghmans et al., Electrochemistry Communications 72 (2016) 113–117)

13.   Finally, the authors should state which concentration is shown in Fig. 9 and explain in more detail how they performed the 90h ED process. What does it mean that the concentration of Cl- and SO42- remains approximately constant? If this represents the concentration in the diluting compartment, I would expect a decrease of concentration over time.

Author Response

Point-to-Point Response to Reviewers' Comments:

ReManuscript ID membranes-435784

Reviewer #2:

Q1. The title of the manuscript is somehow confusing. It gives more importance to the inspiration on mussel secretions than other, in my opinion, more relevant aspects of the work. It does not express which kind of permselectivity is improved (for monovalent anions) and it leads to the idea that the antibacterial property is caused by the “mussel inspired surface functionalization” instead of silver reduced on the membranes.

R: Thank you for pointing out our carelessness. We have explained the permselectivity in title. The preparation of membrane was based on the dopamine polymerization and the modification of polydopamine. And the in situ reduction of silver was attributed to the oxidation of catechol to catecholquinone, so the antibacterial property was indirectly endowed by mussel inspired surface functionalization.

Q2. The article should be revised by an English native speaker. Here I report some of the parts in the text which are confusing:

-          Abstract needs exhaustive corrections: PDA coating was fabricated, Michael addition reaction was proceed, diluted department, the method described in this work making… (no verb in this sentence)

-          l65. Would easily attracts bacteria […] Killing the bacterial

-          l130. Riching

-          l215. dealing with glutaraldehyde solution…

-          (l256) it could be explained by the deposited…

-          (l261) the in situ synthesized Ag NPs was endowed the membrane…,

-          (l266) due to the electrostatic charge neutralization between sulfonyl groups,

-          (l274) anions can effectively entering into membrane by Donnan…,

R: Thank you for pointing out our deficiency. We have carefully checked the English writing.

Q3. As mentioned before, the authors should clearly emphasize which type of permselectivity was improved: the readers could think that they refer to the selectivity towards anions, as compared to cations.

R: The extra explanation of permselectivity was added in the title.

Q4. Why do the authors focus on antibacterial growth? Are problems reported with ED processes? Could the authors support the motivation of this type of modification with references reporting problems of anion-exchange membranes with bacteria?

R: Bacterial contamination on membrane surface is very common in practical use, even during storage and transportation. And E. coli can sometimes be seen on the surface of a non-antibacterial membrane after storage in our own laboratory for a period of time under an electron microscope. However, studies on antibacterial properties are more common of pressure-driven membranes, so it drove me to prepare antibacterial anion exchange membranes.

Q5. L 92. What do the authors mean by improved salt rejection? Anion-exchange membranes are expected to let ions pass through, while avoid, at certain level the transport of water. If the transport of Cl- ions is promoted, salt rejection should not decrease.

R: It means that sulfate ion is more difficult to pass through the membrane than chloride ion after membrane modification. The salt rejection of sulfate ion is improved instead of the promotion of the transport of Cl-.

Q6. Coating with Ag has been widely used to promote antibacterial properties in membranes. However, it can also become a critical issue in case Ag is not properly immobilized and it is released, leading to toxic effects on consumers. Did the authors check the stability of Ag coatings?

R: The reduced silver atoms could be bond on the N-site and O-site in polydopamine. And all the modified membranes have been flushed with deionized water several times to make sure that what remains is stable. In addition, silverware is common in daily life, even silver tableware. The toxicity of silver metal is negligible to human body.

Q7. Coating with Ag is implemented to endow the membranes with antibacterial properties, while DMA coating is applied to increase monovalent anion permselectivity. Could the authors explain in more detail which is the purpose of coating with polydopamine? I guess this is an important point, according to the reference to the mussel inspired modification mentioned in the title. However, from reading the text it is not clearly explained.

R: The polydopamine coating supply a stable media layer for post modification. The Ag ions in solution was firstly chelated by the hydroxyl groups on the polydopamine, and then accepted the released electrons, which was produced by the oxidation of catechol to catecholquione, to realize the in situ reduction of coupled Ag ions on the membrane surface.

Q8. Which constant current was applied? Was it below the corresponding limiting current?

R: The applied current during membrane surface resistance test was 0.04 A (5.67 mA/cm2) which was below the limiting current. The limiting current in this system is about 35.4 mA/cm2.

Q9. L224. When the authors say, “the characteristic peak around 1022 cm-2 corresponds to the symmetric stretching…”. Do they mean a negative peak?

R: Yes, in FTIR figure, the characteristic peak means the negative peak.

Q10. On line 261, I do not understand the increase in tunnel conduction effect and field emission conductive effect. Is the ionic conductivity taking place through the ED cell (electrolyte + membrane) of the same character of electric conductivity? How are both types of conductivities related? Is there any reaction occurring in the Ag NPs deposited on the membrane? As far as I know ion exchange membranes are required to have high ionic conductivity but, unless they are used as electrodes, they are not required to have electric conductivity.

R: I have corrected the explanation of the increased conductivity. The reduced silver atoms could be bond on the N-site and O-site in polydopamine layer, which means there were less reactive sites left for DSA. Compared with DA/DSA membrane, the less grafted DSA around the Ag nanoparticles on DA/Ag/DSA membrane surface resulted in the more leaked chlorine ions pass through the selective layer. In other words, chloride ions were easier to pass through the membrane and expressed low surface resistance.

Q11. Please, explain what diameter of inhibition is. This term may not be familiar to many membrane scientists.

R: Bacterial inhibition zone is an area in which bacteria is difficult to grow, under the influence of the antibacterial materials. The diameter of inhibition often indicates that the antibacterial property of the materials.

Q12. It is reported that the successive coatings induce an increase in resistance of the membranes. However, the resistance was calculated using NaCl solutions, without divalent ions. It has been reported in previous studies, that an increase in selectivity for monovalent ions occurs at the cost of an increase in electrical resistance of the membranes. However, the increase in resistance is usually higher when treating solutions containing divalent ions. (Check for example F. Roghmans et al., Electrochemistry Communications 72 (2016) 113–117)

R: It is true that the membrane surface resistance is higher in multivalent ion systems.

Q13. Finally, the authors should state which concentration is shown in Fig. 9 and explain in more detail how they performed the 90 h ED process. What does it mean that the concentration of Cl- and SO42- remains approximately constant? If this represents the concentration in the diluting compartment, I would expect a decrease of concentration over time.

R: The solution on the side of the diluted chamber is exchanged with new feed solution (0.05 M Cl-/SO42-) every two hours and the samples were taken and analyzed for anion content at the end of every two hours. So the stable concentration of Cl- and SO42- indicated the duration performance of modified membranes.

Reviewer 3 Report

This paper reports the results on fabrication and characterization of a novel anion-exchange membrane, which differs from the pristine membrane by antibacterial resistance and monovalent selectivity. Such membranes present a high interest for the practice of electrodialysis because of increasing applications of this method for the treatment of raw natural waters and waste waters where monovalent selectivity is needed to avoid the concentration of bivalent anions and cations and to reduce bio-fouling. The antibacterial property was achieved by synthesizing Ag nanoparticles in situ on the membrane surface.  Monovalent permselective layer was obtained via deposition of PDA layer and introduction of sulfo groups on the surface. Electrodialysis was used to characterize the membrane performance in terms of the permselectivity between Cl and SO42− .

Generally, the methods used for membrane preparation and characterization are adequate; the treatment of experimental results is correct; the manuscript is well written, the sentences are clear. Therefore, the paper may be recommended for publication, however, after a minor revision.

The comments and suggestions are presented below.

Page 1, abstract. Abbreviation PDA must be defined.

 Page 1, line 34: “Anion exchange membrane (AEM) as the core part of electrodialysis system, has the ability to separate anions and cations [9].”   The reference is not really suitable. The fact described in the sentence is too well known.

Page 2, line 48: “Wanget al. [11]”. Please, correct.

Page 3. Data on membrane conductivity in Table 1. The composition and concentration of the solution used for conductivity measurements should be indicated.

Page 4, line 129: “Diaminobenzene sulfonic acid (DSA) which riching of amine functional groups”. Please, revise the sentence.

Page 4, line 154: “effective membrane surface area was 7.065 cm2.”  Page 5, line 181: “area of the membrane is about 19.625 cm”.   See also line 196. It is doubtful that it is possible to measure the membrane surface with such a precision. Please, give more realistic values.

Page 5. Section 2.4.1.

It is known that the method of membrane resistance measurement, which uses direct current, gives a systematic error. There are two sources of this error. First, there is concentration polarization of the membrane, which results in appearance of an additional ohmic resistance [J. Kamcev et al., Journal of Membrane Science 547 (2018) 123]. Second, concentration polarization produces also a difference in concentration (or diffusion) potential. There is a contribution of the diffusion potential in the diffusion layers adjacent to the membranes, and a sum of two interfacial Donnan potential drops on both sides of the membrane [Belova et al., J. Phys. Chem. B, Vol. 110, No. 27, 2006].  It is true that many authors apply this method, and when a 0.5 M solution is used, the error is not enormous [J. Kamcev et al., Journal of Membrane Science 547 (2018) 123] , however, I recommend to the author to mention possible errors in this paper.

Page 6, line 191: Instead of “the transport number of the ions through the membrane” , it should be “the transport number of the ions in the membrane”.

Page 6, line 195. Please, make identical notation for the volume in Eq. (4) and its description.

Page 12, line 363: “momvalent”. Correct, please.

Author Response

Point-to-Point Response to Reviewers' Comments:

ReManuscript ID membranes-435784

Reviewer #3:

Q1. Page 1, abstract. Abbreviation PDA must be defined.

R: We have defined abbreviation of PDA in abstract.

Q2. Page 1, line 34: “Anion exchange membrane (AEM) as the core part of electrodialysis system, has the ability to separate anions and cations [9].”   The reference is not really suitable. The fact described in the sentence is too well known.

R: The reference has been deleted.

Q3. Page 2, line 48: “Wanget al. [11]”. Please, correct.

R: We have corrected it.

Q4. Page 3. Data on membrane conductivity in Table 1. The composition and concentration of the solution used for conductivity measurements should be indicated.

R: Table 1 shows the official membrane parameters. And the composition and concentration of the solution used for conductivity measurements was 0.5 M NaCl.

Q5. Page 4, line 129: “Diaminobenzene sulfonic acid (DSA) which riching of amine functional groups”. Please, revise the sentence.

R: The sentence has been revised.

Q6. Page 4, line 154: “effective membrane surface area was 7.065 cm2.”  Page 5, line 181: “area of the membrane is about 19.625 cm”.   See also line 196. It is doubtful that it is possible to measure the membrane surface with such a precision. Please, give more realistic values.

R: The effective area of the membrane is determined by the diameter of the inner ring of the self-made device, the device is customrized to this specification.

Q7. Page 5. Section 2.4.1.

It is known that the method of membrane resistance measurement, which uses direct current, gives a systematic error. There are two sources of this error. First, there is concentration polarization of the membrane, which results in appearance of an additional ohmic resistance [J. Kamcev et al.,Journal of Membrane Science 547 (2018) 123]. Second, concentration polarization produces also a difference in concentration (or diffusion) potential. There is a contribution of the diffusion potential in the diffusion layers adjacent to the membranes, and a sum of two interfacial Donnan potential drops on both sides of the membrane [Belova et al., J. Phys. Chem. B, Vol. 110, No. 27, 2006].  It is true that many authors apply this method, and when a 0.5 M solution is used, the error is not enormous [J. Kamcev et al., Journal of Membrane Science 547 (2018) 123] , however, I recommend to the author to mention possible errors in this paper.

R: In order to reduce the concentration polarization and diffusion effect of the solution in the feed chamber, the solutions in the two chambers are interlinked and the continuous flow is induced by a pump during resistance measurement process.

Q8. Page 6, line 191: Instead of “the transport number of the ions through the membrane” , it should be “the transport number of the ions in the membrane”.

R: We have corrected it.

Q9. Page 6, line 195. Please, make identical notation for the volume in Eq. (4) and its description.

R: The description of volume has been corrected in Eq. (4).

Q10. Page 12, line 363: “momvalent”. Correct, please.

R: Thank you for pointing out our carelessness.

Reviewer 4 Report

The main problem of the submitted manuscript is very poor English and too concise description of both used methods and obtained results.

I am not a native speaker but apart from the fact that the grammar is very poor, the manuscript contains a relatively large number of sentences which do not make much sense or even seem contradictory to what one expects.

Below I add my specific comment:

title: I would not use mussel's inspired in the title. It is not the authors' idea, they simply employed what someone else has developed. Moreover, there is no mention of what exactly is "mussels inspired" throughout the text. I would welcome a paragraph about that in the introduction section.

Table1: Add the fixed charges (functional groups) in the used membranes. They are important in explaining the changes in zeta potential when the membranes are exposed to solutions with various pH.

There are many abbreviations (even in the abstract) without explanation what they stand for.

Experimental section:

I would welcome a schematic of the reactions showing the step-by-step chemical modification of the membranes. The main emphasis should be put ont the functional groups introduced on the surface of the membranes.

Equation 1: How U0 was obtained? Was there any flow applied through the chambers of the cell? Was the current density small enough to measure the resistances in the underlimiting region? What is the limiting current density for the studied system?

How was the titration (determination of IEC) exactly carried out? If I only pour silver nitrate into NaCl, I will not know when to stop. In eqn. 2, where does the nH+ come from? How do I calculate its value?

Figure 4 contains schematic of a cell for the permselectivity studies. It only contains 3 membranes (usually five membranes are used) which means that the products of electrochemical reactions taking place on the electrodes can be transported into the desalination chamber. Did the authors observe his effect?

Eqn. 4 should be written in a more elegant way.

Result section:

section 3.1: The authors did not carry out any negative controls. What happens if I do not use DA? Will I still have silver nanoparticles or DSA on the membrane?

section 3.2: Could the authors elaborate on how Tunel conduction effect and Field emission conductive effect can occur on Ag nanoparticles in their system and explain their contribution to the overall smaller resistance of the respective membranes?

section 3.3: Zeta potential is strongly dependent on the pH of the surrounding electrolyte for all the membranes. The surface charge even changes its sign. Could the authors explain what exactly can be happening on the membranes (What functional groups will be responsible for such behavior)?

section 3.4: In fig. 9 the authors show stability of the modified membranes in a long-term electrodialysis. Could the authors also show the surface analysis of the used membranes and compare it to the freshly modified ones?

Author Response

Point-to-Point Response to Reviewers' Comments:

Re:Manuscript ID membranes-435784

Reviewer #4:

Q1. title: I would not use mussel's inspired in the title. It is not the authors' idea, they simply employed what someone else has developed. Moreover, there is no mention of what exactly is "mussels inspired" throughout the text. I would welcome a paragraph about that in the introduction section.

R: Mussel inspiration refers to the self-aggregation and adhesion of dopamine in this article. The post reduction of silver ions and organic reaction were based on the polydopamine layer, which utilized the functional groups in polydopamine.

Q2. Table1: Add the fixed charges (functional groups) in the used membranes. They are important in explaining the changes in zeta potential when the membranes are exposed to solutions with various pH.

R: We have added the functional groups.

Q3. There are many abbreviations (even in the abstract) without explanation what they stand for.

R: We have added the explanation of these abbreviations mentioned in this article.

Q4. I would welcome a schematic of the reactions showing the step-by-step chemical modification of the membranes. The main emphasis should be put ont the functional groups introduced on the surface of the membranes.

R: The schematic of the membrane modification has been added in graphic abstract.

Graphic Abstract: Illustration of membrane surface modification.

Q5. Equation 1: How U0 was obtained? Was there any flow applied through the chambers of the cell? Was the current density small enough to measure the resistances in the underlimiting region? What is the limiting current density for the studied system?

R: U0 was measured in the absence of membrane. The minimum limiting current density of the studied system was 35.4 mA/cm2 which was much higher than test current density 5.67 mA/cm2. The flow rate of solution used during the test was 150 mL/min.

Q6. How was the titration (determination of IEC) exactly carried out? If I only pour silver nitrate into NaCl, I will not know when to stop. In eqn. 2, where does the nH+ come from? How do I calculate its value?

R: Thank you for pointing out our mistake, the H+ in equation 2 should be replaced by Cl- in this case. The titration was carried out via Automatic Potentiometric Titrator (METTLER TOLEDO T50, Switzerland). This is a fully automatic titration instrument, the end point of titration is indicated by a sudden jump of electrode potential.

Q7. Figure 4 contains schematic of a cell for the permselectivity studies. It only contains 3 membranes (usually five membranes are used) which means that the products of electrochemical reactions taking place on the electrodes can be transported into the desalination chamber. Did the authors observe his effect?

R: A low current density was used during the test procedure to avoiding the electrode reaction. And sodium sulfate solution was used as the electrode chamber, even if the electrode reaction occurs, it will not affect the ions in the diluted chamber.

Q8. Eqn. 4 should be written in a more elegant way.

R: Thank you for your suggestion.

Q9. section 3.1: The authors did not carry out any negative controls. What happens if I do not use DA? Will I still have silver nanoparticles or DSA on the membrane?

R: The polydopamine layer supplied a media layer which helped reduce the silver ions in solution and provide reactive sites, as shown in Graphic abstract. This is the foundation of the modification.

Q10. section 3.2: Could the authors elaborate on how Tunel conduction effect and Field emission conductive effect can occur on Ag nanoparticles in their system and explain their contribution to the overall smaller resistance of the respective membranes?

R: Sorry to make such an inappropriate explanation. We have corrected it with more reasonable explanation. The reduced silver atoms could be bond on the N-site and O-site in polydopamine layer, which means there were less reactive sites left for DSA. Compared with DA/DSA membrane, the less grafted DSA around the Ag nanoparticles on DA/Ag/DSA membrane surface resulted in the more leaked chlorine ions pass through the selective layer. In other words, chloride ions were easier to pass through the membrane and expressed low surface resistance.

Q11. section 3.3: Zeta potential is strongly dependent on the pH of the surrounding electrolyte for all the membranes. The surface charge even changes its sign. Could the authors explain what exactly can be happening on the membranes (What functional groups will be responsible for such behavior)?

R: The amount of surface charge depends on the pH of the solution, and the zeta potential will be affected by the surface material. The membrane surface containing sulfonic groups will gain more negative electricity as ph increase, and will tend to be positively with the decrease of pH. In this article, the amino groups on membrane surface would be protonated with pH decrease, which would express the positive charged. The phenolic hydroxyl would be deprotonated with pH increase and express the negative charged. So this is the result affected with both various groups on membrane surface and changing pH.

Q12 section 3.4: In fig. 9 the authors show stability of the modified membranes in a long-term electrodialysis. Could the authors also show the surface analysis of the used membranes and compare it to the freshly modified ones?

R: There is no obvious change in the composition and morphology of the membrane surface. 

Round 2

Reviewer 2 Report

The authors have improved some parts of the manuscript. They changed the title of the manuscript, clarified on what consisted the 90-hour electrodialysis, and slightly corrected the grammar. Nonetheless, the article still needs to be improved to be accepted:

Q1: The article still needs to be proof-read by a native English speaker. Just in the abstract I can detect a sentence which makes no sense "Turned out that the excellent durability of the modified layer on membrane surface, the concentration of Cl- and SO42- in diluted chamber fluctuated around 0.024 and 0.03 mol.l-1 with no distinct decline"

Q5: SO42- is an anion, not a salt. The term improved salt rejection is still incorrect for ion-exchange membranes. A salt is formed by a cation and an anion, and ion-exchange membranes are selective either for anions or for cations, so one cannot write about salt rejection, unless it is referred to an ED stack considering the combined transport through anion-exchange and cation-exchange membranes.

Q8: Authors should not only answer the queries of the reviewers, but also include the information in the manuscript. If reviewers had some questions, probably future readers will have them again if the necessary information is not incorporated in the manuscript.  Now, in the answer I read that a current of 5.6 mA/cm2 was applied, while in the manuscript one reads 5.1 mA/cm2. The same would apply for other questions raised during the first review of the article.

Q12: If it is true that the membrane resistance is higher with solutions containing multivalent metals, it should be indicated in the manuscript. The conduction of such experiments is fast and easy, just with three of the membranes and the virgin ones in a mixture containing sulfate and chloride ions. Otherwise, the impression given from Table 2 is that the increase in resistance can be neglected, whereas in theory, these membranes are synthesized for working with multivalent ions.

Author Response

Point-to-Point Response to Reviewers' Comments:

Re:Manuscript ID membranes-435784

Reviewer #2:

Q1. The article still needs to be proof-read by a native English speaker. Just in the abstract I can detect a sentence which makes no sense "Turned out that the excellent durability of the modified layer on membrane surface, the concentration of Cl- and SO42- in diluted chamber fluctuated around 0.024 and 0.03 mol•l-1 with no distinct decline"

R: We have amended the sentence in the abstract.

Q2. SO42- is an anion, not a salt. The term improved salt rejection is still incorrect for ion exchange membranes. A salt is formed by a cation and an anion, and ion-exchange

membranes are selective either for anions or for cations, so one cannot write about salt rejection, unless it is referred to an ED stack considering the combined transport through anion-exchange and cation-exchange membranes.

R: Thank you for pointing out our mistake. We have corrected it in revised manuscript.

Q3. Authors should not only answer the queries of the reviewers, but also include the information in the manuscript. If reviewers had some questions, probably future readers will have them again if the necessary information is not incorporated in the manuscript. Now, in the answer I read that a current of 5.6 mA/cm2 was applied, while in the manuscript one reads 5.1 mA/cm2. The same would apply for other questions raised during the first review of the article.

R: The 5.6 mA/cm2 was used in the resistance test system, while the 5.1 mA/cm2 was used in the selective test system. And the test system we used is consistent with our previous research. It is easy to compare with the results of our previous studies.

Q4. If it is true that the membrane resistance is higher with solutions containing multivalent metals, it should be indicated in the manuscript. The conduction of such experiments is fast and easy, just with three of the membranes and the virgin ones in a mixture containing sulfate and chloride ions. Otherwise, the impression given from Table 2 is that the increase in resistance can be neglected, whereas in theory, these membranes are synthesized for working with multivalent ions.

R: We have supplemented the test of membrane resistance in system of multivalent ions.

Reviewer 4 Report

I would really recommend to check the English grammar.

Author Response

Point-to-Point Response to Reviewers' Comments:

ReManuscript ID membranes-435784

Reviewer #4:

Thank you for your suggestion, the grammar of manuscript has been further checked.

Round 3

Reviewer 2 Report

The authors have addressed the questions raised during the review. Now data presented in the article has gained in clarity significantly. I still would recommend a last proof-reading by an English native speaker, since I detected some ortographic errors.

Author Response

Point-to-Point Response to Reviewers' Comments:

Re:Manuscript ID membranes-435784

Reviewer #2:

We have further checked and revised the English writing of this manuscript.
